# Active BIM Approach to Optimize Work Facilities and Tower Crane Locations on Construction Sites with Repetitive Operations

**Borna Dasović** [1], **Mario Galić** [2,*] 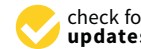 **and Uroš Klanšek** [3]

[1] Josip Juraj Strossmayer University of Osijek, Faculty of Civil Engineering and Architecture Osijek, Vladimira Preloga 3, 31000 Osijek, Croatia; borna.dasovic@gmail.com
[2] Josip Juraj Strossmayer University of Osijek, Faculty of Civil Engineering and Architecture Osijek, Vladimira Preloga 3, 31000 Osijek, Croatia
[3] University of Maribor, Faculty of Civil Engineering, Transportation Engineering and Architecture, Smetanova 17, 2000 Maribor, Slovenia; uros.klansek@um.si
* Correspondence: mgalic@gfos.hr; Tel.: +385-31-540-091

**Abstract:** This paper presents an active building information modeling (BIM) approach for work facilities and the optimal positioning of tower cranes on construction sites with repetitive operations. In this context, the metamorphosis of a passive BIM approach into an active approach is described. Here, the enhancement of the construction-ready BIM model starts with the export of the optimization input parameters, such as the 3D coordinates of the building, perimeter of the construction site, space for feasible solutions, relevant segment of the building with repetitive works, etc. Depending on the complexity of the problem, the user selects a suitable optimization approach and formulates the tower crane positioning optimization problem with the objective of minimizing the total duration of the operation's cycle. Similarly, according to the model formulation, the user also chooses the optimization tool, including the search algorithm. The final step involves the post-optimal analysis and importing of the optimal solution into the BIM. An application example is demonstrated at the end of the paper to show the advantages of the proposed approach in which the optimization model has significantly improved the initial solution of the crane and depot positions.

**Keywords:** active BIM; optimization; positioning; time cycle; tower crane

---

## 1. Introduction

Positioning a tower crane on a construction site is a serious task, regardless of the characteristics of the crane or construction site [1–4]. The importance of the positioning of the tower crane is not only related to the overall safety, but also to its efficiency, which corresponds to its justifiability in terms of economics and energy consumption. By rationalizing the crane's operational time-cycles, its economic efficiency and environmental footprint are rationalized as well. Such conclusions have been presented in previous research. For example, in [2], the authors analyzed the optimal positioning of a tower crane with regard to its type, productivity, and CO2 emissions. In [3], the author claimed that the total electricity requirement for a construction site utilizing even a common electrical tower crane can increase up to 44%. Furthermore, in [4], the authors provide a concise review of the organizational challenges in construction projects involving multiple tower cranes on a construction site. In [5], the authors presented a scenario simulation approach to ensure a lean and environmentally acceptable supply chain on a construction site where the relevant segments in the supply chain were tower cranes powered by different energy sources.

In practice, a common decision-making process regarding tower crane positioning is mainly based on practitioners' intuition and experience. Hence, the position of a tower crane can hardly be defined and confirmed as optimal. Building information modeling (BIM) offers a valid spectrum of information needed for the optimal positioning of a tower crane. However, BIM is not necessarily an optimization tool. The main downside of optimization techniques is considered to be their high mathematical orientation, and in most cases the results have to be interpreted separately, because they tend to be difficult to understand for most project stakeholders.

A dynamic system connecting optimization techniques and BIM is considered an "active BIM". The authors in [6] have reported that active BIM represents an enriched BIM system with additional decision-making functions in order to provide active solutions, while the authors in [7] were more precise, saying that "active BIM functions visually comprehend a problematic situation, and use methodology to present the improved situation". In recent studies, various authors have suggested rather sophisticated and complex approaches for combining optimization techniques and BIM, whilst upgrading BIM to active BIM to solve the location of tower cranes on construction sites.

In this paper, the authors present an active BIM approach for the optimal simultaneous positioning of tower cranes and facilities on construction sites in case of repetitive works, which is a more challenging task. After the introductory chapter, the second chapter contains recent similar approaches, their outcomes, and shortcomings. In chapter three, the authors present the data export methodology from BIM and how it is imported into the optimization model. In chapter four, the application of the approach is presented, followed by chapter five with the interpretation of the results. At the end of the paper, the authors provide discussion and conclusions regarding the presented approach.

## 2. A Short Overview of Recent Active BIM Approaches, Their Outcomes, and Shortcomings

The optimal positioning of tower cranes, with the objective of minimizing the total operation time-cycles of the cranes, is a relevant scientific topic. The importance of the task stems from the commonly known facts that tower cranes are significant electricity consumers on construction sites, and that the electricity price is continuously increasing in most countries in the world. Over the past few decades, various optimization models have been structured and verified to solve such problems, whilst in the past decade, BIM has emerged as suitable for its conjunction with optimization methods. Theoretically, this combination could provide a synergy of those approaches. On the one hand, BIM could dynamically gain verified (i.e., optimal) information, while on the other hand the results of the optimization can be simulated along with other information in the project.

BIM is a dynamic system of project information. However, in terms of engaging optimization methods, BIM can be passive or active. Active BIM engages optimization methods for quantitative mathematical analysis and the confirmation of project information. Even though those two concepts can hardly be applied to the same project without directly combining them, there is not as much anticipated published research on the active BIM topic. In the following text, the authors provide a short review of recently (i.e., from the last decade) published journal articles presenting active BIM approaches for solving the optimal positioning of tower cranes on construction sites.

One of the earliest published approaches of active BIM is from 2012 [8], and combines GIS and BIM with an optimization algorithm, for the optimal positioning of tower cranes on construction sites by minimizing potential conflicts between cranes and facilities on site. The main drawback of their approach was the limitations in integrating GIS and BIM. Apart from the shortcomings of the model noted by the authors, the applied optimization technique and tool are not clearly stated in the paper nor is its relationship with BIM, which makes it hard to draw any general conclusions. In 2013, the same authors extended their approach to supply chain management, which may be useful for optimizing crane operation [9].

In [10], the authors presented an active BIM approach using the firefly algorithm (FA) for the layout optimization of the tower crane with a similar optimization objective as in the previous case. It was concluded that the model was applicable and provided a solution much more quickly than

previous methods to solve the layout in case of multiple tower cranes on site. Visualizations of the results are also much easier to present to field workers. A similar approach was presented in [11], but with a slightly different objective function of the optimization (i.e., minimization of the number and optimal location of tower cranes). However, in both methods, the main disadvantage was that FA, as with most heuristics, is sensitive in terms of the local suboptimal solutions, or "premature solutions" as was elaborated in the paper [12].

A BIM-based optimization model for tower crane selection, their number, and layout was presented in paper [13]. The authors used the analytical hierarchy process (AHP) for the tower crane type selection, and a genetic algorithm (GA) was used for determining the optimal number and layout of the cranes. The model was applied to a case study problem and yielded good results. As a potential further development, the authors pointed out that the model should be more adjustable to the dynamic nature of construction sites. It is also known that the AHP method by default is not an optimization method, and the weights of the criteria are susceptible to the subjective volition of the user.

A mixed integer programming optimization model for optimizing tower cranes and allocating material supply points on construction site was presented in [14], with the objective to minimize operational and rental costs. However, the optimization results merged with BIM were not explicitly presented, but were rather presented in such a manner that the presented model was a BIM-ready optimization approach. Another active BIM approach was used for solving tower crane location, presented in [15]. The authors exported the construction site layout from BIM as a binary image file into MATLAB, where the optimization was carried out. Unfortunately, as in the previous case, explicit mathematical definitions of the constraints and objective function were not presented in the paper.

An interesting model for solving the location of a tower crane on a construction site, even though it did not engage any optimization method or tool, is presented in [16]. The model offers a subset of feasible solutions that should be applied to a suitable optimization method, which the authors underlined as the right way of developing their model.

A BIM-based approach to construction site facility optimization is presented in [17]. The optimization objective in the model was to determine temporary facility layouts that would minimize onsite transportation costs, whilst not compromising the safety or accessibility of the site. The tower crane was included as a reachability constraint. For the optimization, the authors used GA. Similar studies and active BIM-ready approaches were presented in [18–20], while in [21,22] the BIM's potentials in terms of its further development were presented. A more extensive review of applied active BIM approaches, not only those concerned with tower crane or construction site layout optimization, is given in [23].

The optimization approach, which was the motivation for the study presented in this paper, is provided in [24]. The model presented by the authors in this paper did not use BIM, but rather a 2D computer-aided design (CAD) model for determining the optimal location of a tower crane. Besides this, the optimization model was GA-based and it was oriented only to positioning the tower crane, while the locations of the site facilities were fixed.

In the model presented in this paper, the authors extended the optimization model to the optimal positioning of multiple work facilities on a construction site as well as positioning a single tower crane. The presented approach required a construction-ready BIM model with a digital dataset for the optimization process. Besides the spatial and physical constraints on feasible solutions for positioning the tower crane and work facilities, BIM provided specific information for the optimization process, (e.g., a work breakdown structure of the building, and the types and nature of the material and elements required, etc.). The information obtained by the optimization process was transferred back into the BIM model, which enhanced it to an active BIM system through its two-directional, dynamic, and adjustable connection with the optimization model.

## 3. Active BIM Approach for the Optimal Positioning of Tower Cranes and Facilities

### 3.1. Methodology of Structuring the Input Parameters Exported from BIM

BIM is a complex system of information compressed into a digital model. The level of detail (LOD) is a pragmatic structure of labels that define what and where users can expect and use from a BIM model. However, the details and information for modeling a construction site layout differ significantly from the details needed for building. In the literature, this approach of BIM LOD is known as the ergo-technique design [25], where information is intended for a pre-construction contractors' use. In [26], it was underlined that, among others, the linked data approach with semi-automated data transfer is the most promising.

In this context, the authors present a general process flow metamorphosis of BIM into an active BIM approach for solving the simultaneous positioning of tower cranes and work facilities. As shown in Figure 1, the enhancement of the construction-ready BIM model starts with the export of optimization input parameters (i.e., 3D coordinates of the building, the perimeter of the construction site, the space for feasible solutions, and relevant segments of the building with repetitive works). To ensure a dynamic system for the connection between the BIM environment and optimization model environment, it is important that the script of input parameters is transferred from the BIM to a document with a compatible extension to the optimization model environment, in this case, a spreadsheet (e.g., MS Excel format). Depending on the complexity of the problem, the user determines a suitable optimization approach and models the optimization problem of the tower crane positioning with the objective of minimizing the total duration of the operation cycle. Similarly, depending on the problem's complexity and its model formulation, the user chooses the optimization tool and an appropriate optimization algorithm. The final step is the post-optimal analysis and importing of the optimal solution into the BIM, transforming the model into an active BIM model.

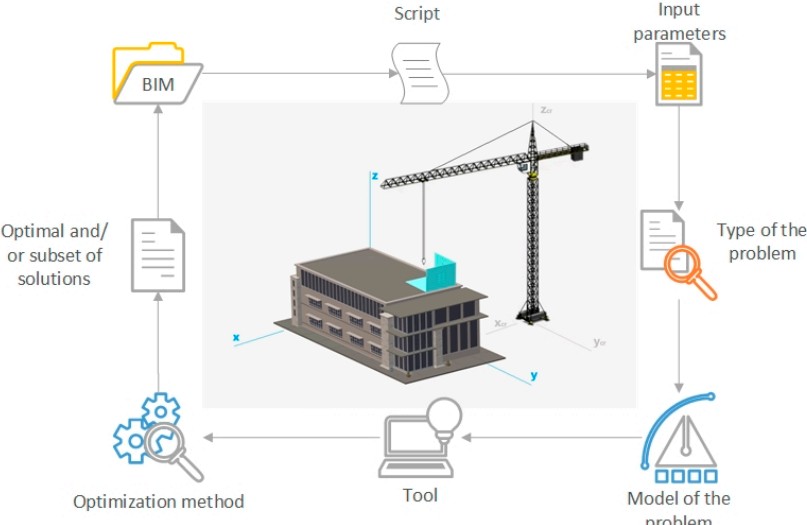

**Figure 1.** Flow diagram of the active building information modeling (BIM) process.

### 3.2. Optimization Model Formulation

The criterion of the optimization determined in the proposed model is to minimize the total duration of all the cycles required to supply materials from storage to the points of demand on the building. Therefore, the objective function is formulated in the following form:

$$\min Z = \sum_{i=1}^{I} \sum_{j=1}^{J} Nc_{i,j} \cdot T_{i,j} \tag{1}$$

where $i$, $i \in I$, denotes the set of storages; $j$, $j \in J$, represents the set of demand points; $Nc_{i,j}$ stands for number of supply cycles between the $i$-th storage and the $j$-th demand point; and $T_{i,j}$ indicates the time of the supply cycle.

The first set of constraints defined by inequalities (2) and (3) assures that the tower crane is located out of the storage areas:

$$|Xcr - Xs_i| - \frac{Lcrx + Lsx_i}{2} \geq 0 \qquad i \in I \tag{2}$$

$$|Ycr - Ys_i| - \frac{Lcry + Lsy_i}{2} \geq 0 \qquad i \in I \tag{3}$$

where ($Xcr$, $Ycr$) are position coordinates of tower crane; ($Xs_i$, $Ys_i$) are position coordinates of the $i$-th storage center; ($Lcrx$, $Lcry$) are the rectangular dimensions of the tower crane foundation area; and ($Lsx_i$, $Lsy_i$) are rectangular dimensions of the $i$-th storage area.

The constraints set by inequalities (4) and (5) ensure that the tower crane is located outside the areas of demand points:

$$|Xcr - Xd_j| - \frac{Lcrx + Ldx_j}{2} \geq 2 \qquad j \in J \tag{4}$$

$$|Ycr - Yd_j| - \frac{Lcry + Ldy_j}{2} \geq 2 \qquad j \in J \tag{5}$$

where ($Xd_j$, $Yd_j$) denote the position coordinates of the $j$-th demand point while ($Ldx_j$, $Ldy_j$) represent rectangular dimensions of the $j$-th demand area.

It was assumed that the storage areas cannot be located in the demand areas and this was provided by inequalities (6) and (7):

$$|Xs_i - Xd_j| - \frac{Lsx_i + Ldx_j}{2} \geq 1 \qquad i \in I \quad j \in J \tag{6}$$

$$|Ys_i - Yd_j| - \frac{Lsy_i + Ldy_j}{2} \geq 1 \qquad i \in I \quad j \in J \tag{7}$$

The overlapping of different storages in the same area was prevented by inequality constraints (8) and (9):

$$|Xs_\mu - Xs_\pi| - \frac{Lsx_\mu + Lsx_\pi}{2} \geq 0 \qquad \mu, \pi \in I; \mu \neq \pi \tag{8}$$

$$|Ys_\mu - Ys_\pi| - \frac{Lsy_\mu + Lsy_\pi}{2} \geq 0 \qquad \mu, \pi \in I; \mu \neq \pi \tag{9}$$

where $\mu$ and $\pi$ represent feasible pairs of different storages contained within the set $i \in I$.

Bounds on the position coordinates of the tower crane, defined by inequalities (10) and (11), ensure that its location is situated within the construction site plot:

$$X^{LO} + \frac{Lcrx}{2} \leq Xcr \leq X^{UP} - \frac{Lcrx}{2} \tag{10}$$

$$Y^{LO} + \frac{Lcry}{2} \leq Ycr \leq Y^{UP} - \frac{Lcry}{2} \tag{11}$$

Similarly, inequalities (12) and (13) were set to assure that the storage of materials was also located inside the construction site plot:

$$X^{LO} + \frac{Lsx_i}{2} \leq Xs_i \leq X^{UP} - \frac{Lsx_i}{2} \qquad i \in I \tag{12}$$

$$Y^{LO} + \frac{Lsy_i}{2} \leq Ys_i \leq Y^{UP} - \frac{Lsy_i}{2} \qquad i \in I \tag{13}$$

All other optimization model entities, such as geometrical conditions, transportation time restraints, and velocity constraints, among others, were formulated as proposed in reference [24]. In this way, the output results provided by the proposed model included the optimal position coordinates of the tower crane, as well as the optimal locations and areas for the storage of materials.

### 3.3. Optimization Problem Solution

The optimization model was computer-generated utilizing the Microsoft Excel (2016) software, while the optimization process was executed with an add-in called Solver, developed by Frontline Systems (2016). From the viewpoint of mathematical modeling, the objective function, as well as the (in)equality constraints of the optimization model, were determined with smooth and differentiable expressions while the output results were defined to be handled by continuous variables.

Therefore, the exact nonlinear programming approach was employed to optimally solve the stated problem. The latest version of Solver's generalized reduced gradient method (GRG), originally proposed in [27], was selected and applied to perform the optimization. With regard to the convergence of the optimal solution, the default termination tolerances were set for the GRG search algorithm to carry out the optimization. The following section is intended to demonstrate the applicability of the proposed approach.

## 4. Application Example

### 4.1. General Information and Input Parameters

An application example will be demonstrated using the actual construction site of a multi-story residential building in Osijek, Croatia. The considered building object consisted of a basement, ground floor, and five floors, with the layouts from first to the fourth floor repeated in both form and wall materials (masonry blocks and mortar). The highest point of the building was +20.22 m above the surrounding terrain. Technical data, along with CAD digital drawings, were obtained from the design studio in charge of the project [28]. For this study, the authors structured a BIM model, shown in Figure 2.

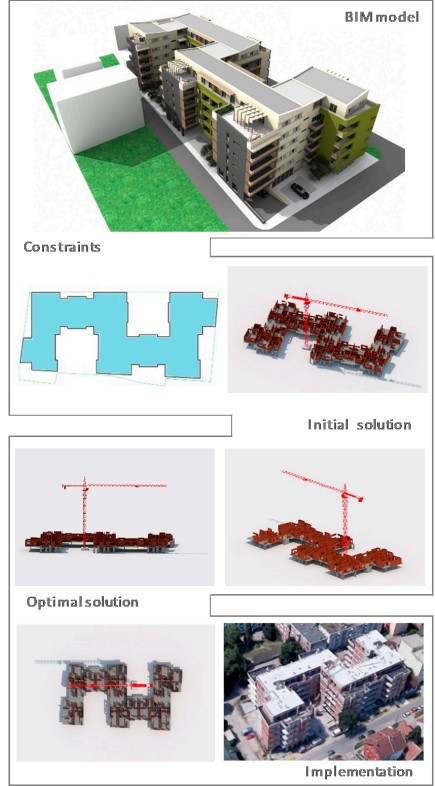

**Figure 2.** Process flow of the applied methodology.

The construction site was located in the narrower city center and bordered on all sides with existing facilities and a one-way road. The building object extended across the entire land plot, thus leaving limited areas for the positioning of the tower crane and material storage. The optimization problem discussed here was to simultaneously identify the optimal position of the pre-selected tower crane and the location of the material storage. For simplicity, this particular example focused only on the storage of the masonry blocks and mortar production plants.

The BIM tool used for this case study was Allplan 2018 [29], which directly exported the required input parameters to the spreadsheet in which the optimization model was structured. The process flow from the BIM model to the optimal solution ready for implementation is shown in Figure 2.

The number of input parameters for the optimization obtained from the BIM was surprisingly small. The input parameters consisted of:

- coordinates, material, and the required quantities of the demand points. In this case, the study materials were the masonry blocks and mortar needed for the construction of the first floor, the layout of which was repeated until the fourth floor (the demand points were the walls with quantities of blocks and mortar that they required. These were structured in 16 demand points in order to keep the variables to a reasonable number),
- coordinates of the feasible tower crane positions. The case study construction site was extremely narrow, which reduced the space for feasible tower crane positions, as well as the positions of the mortar plants (the base of the local coordinate system was set in the far left corner of a land plot),
- number and type of the available tower cranes. Since the case study was an ongoing construction site, the authors used the same tower crane as was used on the construction site (i.e., Liebherr's 132 EC-H8 Litronic tower crane with technical data obtained from [30]) and used the actual crane's position as the initial solution of the optimization. It is necessary to underline here that the crane model used in the BIM did not match the one mentioned earlier. However, the model used fit the purpose in terms of its dimensions and physical characteristics. The model of the crane had a static nature in the BIM model; the model also presents ist spatial and physical positioning. The crane's capacity, with regard to the radius, was modeled as a logical two-part function where the first part is a linear function limited to 8000 kg up to a 15 m radius, while the second part is a cubic polynomial function up to a maximum radius of 55 m when the capacity is 1850 kg. The velocities of the crane's operations were modeled according to the information given in [30], where the factors of the simultaneous operation relationships, suggested by the authors in [28], were taken into account.

### 4.2. Optimization and Results

Optimization was performed on a 64-bit operating system with personal computer, with processor AMD A4-3300M APU (dual-core 1.9 GHz), 4 GB random access memory, and a 620 GB hard drive. After running the Solver's GRG search algorithm, 101.292 seconds of CPU time was required to achieve the convergence of the optimal solution. At this point, the optimal local position coordinates for the tower crane (49.72, 26.55), brick storage (28.38, 13.52), and mortar production plant (47.93, 21.75) were obtained at the minimum total duration of supply cycles, which amounted to 294.80 minutes. In a comparison, the initial position of the crane and material storage generated a total operation cycle of 450.19 minutes, thus the result obtained using the optimization model presents a time saving of 34.7 %.

After this, the local position coordinates were transformed into global coordinates according to the Croatian reference coordinate system for map projection HTRS96/TM [31] as follows: Tower crane (East: 669341.63, North: 5048795.89), brick storage (East: 669317.06, North: 5048791.25), and mortar production plant (East: 669338.26, North: 5048792.04). It was therefore possible to perform on-site positioning. The optimal positioning of the tower crane, brick storage, and mortar production plant is shown in Figure 3.

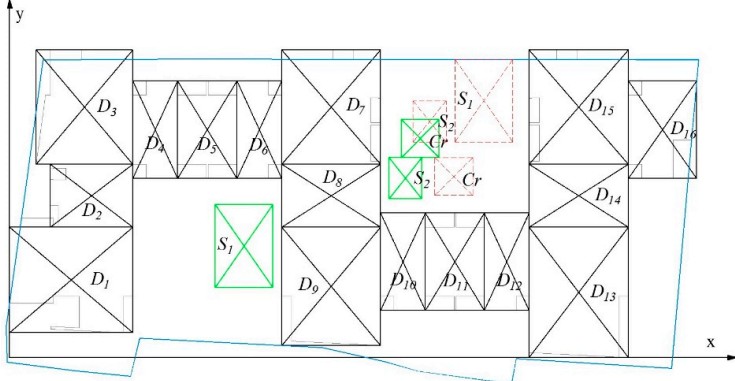

**Figure 3.** Optimal positioning of the tower crane, brick storage, and mortar production plant.

Figure 3 demonstrates that all optimization constraints were fulfilled, that is, there was no overlap between areas related to the tower crane, brick storage, and mortar production plant (the initial positioning was indicated with red lines while the optimal positioning was represented with green ones). Figure 3 shows that the optimal positioning was achieved within the land plot while the material sources and the building object were in the range of the tower crane.

## 5. Discussion and Conclusions

This paper presented the active BIM approach for the optimal simultaneous positioning of a tower crane and the facilities required to support the execution of repetitive works on a construction site. The output results produced using the proposed model include the optimal position coordinates of the tower crane, as well as the optimal locations and areas of the material storage.

The application example demonstrated that using the active BIM approach can achieve time and cost savings in terms of material transport. Although model management requires some effort when dealing with decision variables, an advantage that can be highlighted here is that the optimization modeling and solving were performed within the well-known and user-friendly environment of Microsoft Excel, which is widely used in construction practice. The total crane operation time savings due to the definition of the crane's optimal position was 34.7 %. The effort needed to achieve this crane efficiency improvement was rather minimal and requires basic engineering knowledge in programming and practice. This was the premise of this research, i.e., to present an efficient and simple approach for solving the addressed problem. To the best of our knowledge, this is the first time that a decision-support system based on the methodology proposed in this paper and applied to an actual example from construction practice has been presented.

When further developing the approach, it is advisable to include a dynamic crane in the active BIM, as this means that the visualization of the crane's operation could help to resolve eventual clashes and might improve the solution. In addition, the main limitation to the applicability of the presented model is its application for single crane positioning problems. In a further development, the modification of the optimization model should be expanded be applicable to solving problems with multiple tower cranes.

**Author Contributions:** Conceptualization, M.G. and U.K.; Data curation, B.D. and M.G.; Formal analysis, M.G. and U.K.; Funding acquisition, M.G.; Investigation, M.G.; Methodology, M.G. and U.K.; Project administration, M.G. and U.K.; Resources, M.G.; Software, B.D. and M.G.; Supervision, M.G. and U.K.; Validation, M.G. and U.K.; Visualization, B.D. and M.G.; Writing—original draft, B.D. and M.G.; Writing—review and editing, M.G. and U.K.

**Funding:** The APC was funded by the Faculty of Civil Engineering and Architecture Osijek.

**Acknowledgments:** The authors acknowledge the financial support from the Slovenian Research Agency (Research Core Funding No. P2-0129).

**Conflicts of Interest:** The authors declare no conflict of interest. The funders had no role in the design of the study; in the collection, analyses, or interpretation of data; in the writing of the manuscript, or in the decision to publish the results.

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
