# Peer review of "Active BIM Approach to Optimize Work Facilities and Tower Crane Locations on Construction Sites with Repetitive Operations"

_buildings, doi:10.3390/buildings9010021_

Round 1
Reviewer 1 Report
1. The Introduction should be improved. I cannot get any idea about what is active BIM, why it is important after reading the introduction for three times. 2. What is the research gap? why the proposed method can help address the gap? more clear explanation should be provided. 3. What are work facilities? why not focusing on tower crane? 4. Please provide a brief summary after presenting several previous studies? Please highlight the major gaps that will be addressed in this study only. 5. Actually, the location of tower crane should be determined by many factors, not only the estimated transportation time. Therefore, the proposed method may have limited practical value. 6. It is better to add a table showing the source of data. 7. No comparison with existing methods to show the benefits of the method proposed in this study. 8. Some statements need supporting reference, like Line 30, Line 57, etc.
Author Response
Dear Reviewer,
thank you for taking time and providing us with your helpful advisees and suggestions in order to improve our paper. We have devotedly worked to acknowledge all of your comments and suggestions, incorporate them in the paper along with suggestions given by other two reviewers. All changes are marked yellow in the resubmitted paper, while in the document attached we provide a summary of changes explained and referenced.
Best regards,
Authors

Reviewer 2 Report
The Article is good enough, though I always expect, or in other words prefer, less already known things and a more engineering approach.
Author Response
Dear Reviewer,
thank you for taking time and providing us with your helpful comments in order to improve our paper. We have devotedly worked to acknowledge all comments and suggestions given by the reviewers and incorporate them in the paper. All changes are marked yellow in the resubmitted paper.
Best regards,
Authors

Reviewer 3 Report
The article does not indicate the role of the BIM system in calculations, especially - as the authors themselves write - the role of an active BIM. From what the authors write, the only element of BIM is the model of the building itself. Are the analyzes and calculations connected in any way with the building model and the management of the construction? We learn that the input data have been obtained from the BIM model (in a form of a spreadsheet), but how did the calculation data affect design solutions?
The authors also write that they received blueprints. What for, if the BIM model is an electronic dataset?
Then, how can the presented method be related to the location of more than one crane? Does the theoretical model take into account the specificity of the construction site: the size of the plot, its neighborhood and terrain?
The review describes problems resulting from energy consumption (48-56) - what is the significance for optimizing the position of the crane during construction works?
Author Response

(The authors gave the same response as above.)

Round 2
Reviewer 1 Report
Thanks for your responses.